# Revisiting *Schistosoma mansoni* Micro-Exon Gene (MEG) Protein Family: A Tour into Conserved Motifs and Annotation

**DOI:** 10.3390/biom13091275

**Published:** 2023-08-22

**Authors:** Štěpánka Nedvědová, Davide De Stefano, Olivier Walker, Maggy Hologne, Adriana Erica Miele

**Affiliations:** 1UMR 5280 Institute of Analytical Sciences, Université de Lyon, CNRS, Université Claude Bernard Lyon 1, 69100 Villeurbanne, France; stepanka.nedvedova@gmail.com (Š.N.); olivier.walker@univ-lyon1.fr (O.W.); maggy.hologne@univ-lyon1.fr (M.H.); 2Department of Chemistry, Faculty of Agrobiology, Food and Natural Resources, Czech University of Life Sciences, 16500 Prague, Czech Republic; 3Department of Zoology and Fisheries, Center of Infectious Animal Diseases, Czech University of Life Sciences, 16500 Prague, Czech Republic; 4Department of Biochemical Sciences, Sapienza University of Rome, 00185 Rome, Italy

**Keywords:** micro-exon genes (MEG), *Schistosoma mansoni*, phylogeny, gene annotation

## Abstract

Genome sequencing of the human parasite *Schistosoma mansoni* revealed an interesting gene superfamily, called micro-exon gene (*meg*), that encodes secreted MEG proteins. The genes are composed of short exons (3–81 base pairs) regularly interspersed with long introns (up to 5 kbp). This article recollects 35 *S. mansoni* specific *meg* genes that are distributed over 7 autosomes and one pair of sex chromosomes and that code for at least 87 verified MEG proteins. We used various bioinformatics tools to produce an optimal alignment and propose a phylogenetic analysis. This work highlighted intriguing conserved patterns/motifs in the sequences of the highly variable MEG proteins. Based on the analyses, we were able to classify the verified MEG proteins into two subfamilies and to hypothesize their duplication and colonization of all the chromosomes. Together with motif identification, we also proposed to revisit MEGs’ common names and annotation in order to avoid duplication, to help the reproducibility of research results and to avoid possible misunderstandings.

## 1. Introduction

Global efforts for genome sequencing after the human genome project [1,2] have been extremely beneficial to the entire scientific community. The Wellcome Trust Sanger Center, among others, undertook the task of sequencing the genomes of pathogens, in particular parasites, which affect the vast majority of the world’s population [3,4]. Parasites are complex eukaryotes, whose genomes show more resemblance with their hosts than with non-parasitic family members. In the last twenty years, the completion of parasites’ genomes opened the way to study the molecular basis of the diseases caused by these agents and also to find and validate new drug targets. This last point is a critical one since most of the drugs against parasites are old, most of the time with severe side effects, or not any more effective due to resistance. Nevertheless, focused genetics and molecular biology studies had already been carried out before the genomic era; therefore, quite a number of gene products had been annotated and served as landmarks for the recent annotations.

Under many respects, *Schistosoma mansoni*, the widest-spread agent of human schistosomiasis, is a good case study [4,5]. This vector-borne extracellular parasite is a metazoan digenean trematode; it possesses seven autosomes and a pair of sexual chromosomes, with a complex genetic structure including long interspersed sequences, transposon-like sequences, alternative splicing, and gene duplications [5,6,7,8,9].

One class of genes, in particular, has attracted the attention of researchers, the so-called micro-exon genes (MEG). Their discovery happened well before the completion of *S. mansoni* genome sequencing [10,11], although their annotation was completed with the version available on February 2023 on WormBase ParaSite (WBPS17–WS282) [12]. In total, 35 genes are annotated with characteristic alternants of long introns (ranging from 0.1 to 5 kbp) and very short exons, whose length can vary from 6 to 81 base pairs, with a majority of 15 bp long exons [5,12]. The introns’ structure is also quite intriguing and characteristic: their length is shorter towards the 5′ and 3′ ends (ranging from 100 to 500 bp) and longer (up to 4 kbp) in the center. This peculiarity made their automatic inference complicated from a genomic point of view.

However, the last fifteen years have seen plenty of transcriptomics and proteomics studies accumulating evidence of MEG complexity and variability and allowing for their tracing back to the genome [5,11,13,14,15,16].

In this work, we present a systematic study aiming to understand the putative origins and spread over the entire genome of the MEG superfamily of genes. We highlight the presence of conserved motifs, despite the high variability, and we finally suggest a revision of the nomenclature in order to resolve some ambiguities in the literature.

## 2. Materials and Methods

We have interrogated the WormBase ParaSite (WBPS) database [12] over the last two years, because this is the reference database for helminth, and we present the data from the session WS282 (last accession 31 March 2023), searching, within the genome of *Schistosoma mansoni*, for the terms “Micro-Exon Gene”, “MEG”, “antigen 10.3”, “GRAIL”, and “ESP”. For each gene, WBPS presents its structure, the possibility to retrieve the sequence, the position on the chromosome, and a translation identified with (at least) one associated UniProt ID. On the UniProt website [17], each entry is associated with both its WBPS and GenBank^®^ identifiers, when available.

An example of the WBPS entry for MEG 3.2 isoform 1 is given in Figure 1.

In parallel, we have performed exactly the same search on UniProt KnowledgeBase [17] and cross-verified the results. All the protein sequences downloaded from UniProt were passed into psi-BLAST from NCBI [18], restricting the search to *S. mansoni* in order to collect the maximum number of protein sequences annotated as MEG.

Afterward, we performed a trimming, eliminating the duplicated entries, which displayed the same sequence with two different protein accession numbers, one from NCBI and one from UniProt. Whenever this was the case, we arbitrarily kept the UniProt ID only for consistency with the cross-annotation of WBPS. While we were writing this manuscript, on 13 April 2023, a new release of WPBS was published, which included more transcriptomic data and increased the number of putative *megs*, without the associated protein data; therefore, we did not change our verified workflow.

The primary structure alignment of MEG proteins was performed with T-Coffee [19], K-align [20], and MUSCLE [21] on the EBI server [22] (last access on 31 March 2023). All the default parameters were chosen. The alignments were then manually inspected and the one from MUSCLE was finally preferred for subsequent analysis (Appendix A) because the inserted gaps respected the exons’ boundaries; hence, they better respected the biological constraints of alternative splicing.

Phylogenetic trees were built with both Simple phylogeny [23] and PRANK [24] on the EBI server [22] (last access on 31 March 2023). The one from PRANK was more consistent with the gene clustering and also with the type of retrotransposon sequences, which had been found at the boundaries of the *megs* [7,8]; therefore, it was retained and visualized on the iTOL (interactive Tree of Life) server [25].

Emboss on the EBI server [22] was used to put in evidence conserved linear motifs, which were displayed with Weblogo [26].

Primary sequence analysis was performed by the ProtParam tool [27] on the Expasy website [28] (last access on 31 March 2023); the results on calculated molecular weights, isoelectric point, aliphatic index, and GRAVY index are presented in the Appendix A. In this table both the WBPS gene name and the corresponding code from GenBank^®^ (when available) are listed (last access on 18 August 2023).

## 3. Results and Discussion

The latest annotated genome version of *S. mansoni* contains at least 35 unique micro-exon genes (*meg*) with a peculiar structure of 10 to 20 very short exons interspersed with long introns, whose length spans from 100 to almost 5000 base pairs. Unsurprisingly, the automatic annotation was challenging to trace these genes and to recognize them as protein-expressing ones. The short exons are, in the majority of cases, a multiple of three and range from a minimum of 6 bp (i.e., two amino acids) to a maximum of 81 bp. However, in a few cases, one or two exons contain a number of base pairs not divisible by three. In the vast majority, the exons are 15 bp long, thus coding for five amino acids [5,11,12].

These 35 verified protein-expressing genes are interspersed over the seven autosomes and the sexual chromosomes (Figure 2); most are coded on the leading strand and a few in the complementary one, such as *Smp_010550* (uncharacterized/MEG-15), which interestingly codes for one tRNA on the leading strand.

On chromosomes 1, 3 and 5, they happen to be clustered together, reinforcing the hypothesis of their origin *via* gene duplication.

All the chromosomes contain at least one *meg* (chromosomes 2, 4 and the sexual one), while chromosome 3 hosts thirteen distinct *meg*s. Moreover, close to the 5′ of *megs,* three types of (retro)transposable elements have been found, suggesting a spreading *via* gene duplication and transposition and subsequent mutation. This finding was previously corroborated by studying the ratio between non-synonymous over synonymous (dN/dS) mutations in *meg* [7,8,29].

### 3.1. MEG Filiation

In *S. mansoni*, 35 *megs* code for at least 87 verified MEG proteins with a unique UniProt ID, mainly originating from alternatively splicing the central exons. In the past, the gene sequences have been annotated and clustered into 23 families, numbered from 1 to 16 and from 26 to 32. MEG proteins have been found prior to the genome sequencing in the secretions from eggs and adult worms, in both transcriptomics and proteomics studies. At the beginning, they were named “antigen 10.3”, “Egg secreted protein no 15 (ESP15)”, and “Grail”, before their peculiar gene structure was discovered [5,10,11,13,14,30].

We have aligned the 87 verified proteins, whose sequences are given in the Appendix A, and we found that, among the three most used software offered on the EBI tools webpage [22], MUSCLE [21] was the one that reflected the splicing constraints the most. In fact, it inserted the gaps between exons and not inside them (Appendix A); moreover, it aligned the N-termini and the C-termini better than T-coffee or K-align.

Starting from an unknown ancestor, possibly on chromosome 6, a putative (phylogenetic) ontogenetic tree based on sequence similarities was built by PRANK [24] and a clustering in two main groups/clades was highlighted (Figure 3).

It is plausible that a “proto-*meg*” was on chromosome 6, since the first leaves departing from each clade are there. Indeed, MEG-28 and MEG-29 of the red clade and MEG-8 (*Smp_172180.1*) and MEG-32 of the blue clade are on chromosome 6. Of course, this is just a pure hypothesis based on sequence similarity among the orthologous proteins.

The proposed filiation of the red clade, composed of seven gene families, indicates an early jump on chromosome 3 to give rise to the MEG-1 and MEG-3 protein families, which practically colonized the entire chromosome. On the other hand, *meg-9* and *meg*-*31* jumped, respectively, on chromosomes 7 and 1. Based on sequence similarity, an event of gene duplication to generate *meg-3* and *meg-9* prior to splitting into two different chromosomes could be hypothesized, while *meg-31* could well be a filiation of *meg-28*.

Again on chromosome 6, we find the genes coding for MEG-13, MEG-8 (*Smp_172180.1*), MEG-26, MEG-32 of the second clade, the blue one in Figure 3. Therefore, we might speculate that the first “experiments” on increasing protein variation *via* gene duplication and alternative splicing were carried out in chromosome 6 and then pursued on other chromosomes, in particular, no. 3 and no. 1. In fact, the protein coded by *meg-8* on chromosome 6 is close to the proteins of the MEG-2 family of chromosome 3, which then expanded and duplicated. The goal of increasing variability is quite common to parasites and it is usually linked to strategies for achieving host immune evasion [9,11,29,31,32].

*meg-9* on chromosome 7 was “joined” by *meg-15*, whose sequence is similar to the large *meg-2* protein family; hence, we might speculate a jump from chromosome 3 to no. 7 rather than an intra-chromosomal duplication and mutation. The fact that they are not clustered together might corroborate this hypothesis.

Chromosome 3 is particularly interesting since it contains the highest number of *meg* genes and also the majority of egg-secreted MEG proteins (MEG-2, MEG-3, and MEG-1) [5,14,30], as well as the genes with the highest number of spliced isoforms (14 MEG-1 isoforms from *Smp_122630*, Figure 3).

According to genome annotation, *meg-8* (*Smp_163710.1*), *meg-16,* and *meg-6* are the only representatives on chromosomes 2, 4 and the sexual one, respectively. They all belong to the blue clade, but the protein product of *meg-16* shares more similarities with MEG-32, which may suggest an early jump from chromosome 6 to 4, while MEG-6 proteins are closer to the MEG-2 protein family, suggesting a transposition from chromosome 3 towards the sexual chromosome. Moreover, MEG-8 protein shares 50% similarity with MEG-30, suggesting a transposition from chromosome 1 to chromosome 2. It is worth to remind that the mRNA of MEG-6 and MEG-8 are highly present in the eggs and esophageal excreted/secreted transcripts, and that the respective proteins have been found in proteomic studies [13,14,30].

Looking at the tree in Figure 3, we could also speculate that the clade painted in blue was more successful in diversifying the sequences, an aspect that we have tried to highlight with three shades of blue. On the other hand, we could speculatively infer that the clade painted in red was more successful in implementing the alternative splicing to achieve diversity.

### 3.2. Conserved Motifs

Going back to the total alignment in Appendix A, it is clear that there are very few consensus motifs conserved, except the N-terminal signal peptide, which is needed for secretion. We have therefore decided to present this alignment more graphically, by using WebLogo [26] in Figure 4. Apolar residues such as Pro, Phe, Leu are regularly spaced and highly represented. Charged residues, mostly basic Lys, are more conserved in the C-terminal part, while Glu (E) is interspersed at more or less regular intervals of 15–20 aa (taking into account the gaps).

Soon after the signal peptide, all the MEGs present a stretch of hydrophobic and aromatic residues ended by a basic one: FLϕϕFX_6_W*p*(K/H/R). Where X is any residue, *p* is a polar amino acid, and ϕ is a hydrophobic one.

Independently of the clade, it is apparent that MEG proteins possess sticky sequences, which we have verified by calculating the aliphatic index and the GRAVY index with ProtParam [28]. These data are included in the Appendix A.

The stickiness of MEG proteins also appeared from the studies aimed at producing the isolated proteins through protein engineering. For example, recombinant MEG-3.2 and MEG-3.4 were purified from the inclusion bodies of *Escherichia coli* Rosetta cells and refolded by dialysis after purification, before they could be used for immunization studies [33]. MEG-14 was poorly expressed in bacteria and its amphipathic character was studied using circular dichroism with synchrotron light [34] before using the protein for binding studies with host factors [35]. MEG-4 (Sm10.3) was expressed in *Escherichia coli* Rosetta cells and purified in a buffer containing 0.5 M NaCl [36], a non-physiological hypertonic concentration of salt. MEG-24, MEG-27 and MEG-2.1 could not be produced in a heterologous host and, given their short size, they were chemically synthesized to perform in vitro studies [37,38]. Chemical synthesis was also employed to use peptides of several MEG proteins as baits in search for host partners; this was the case, for example, of MEG-12 [32], MEG-8 [39], and twelve other MEG proteins expressed in the tegument and esophageal glands [40].

If we split the two clades and align the sequences separately (35 proteins for the red clade and 52 for the blue clade), we can appreciate that the contribution of conserved Cys and Phe to the overall alignment comes from the red clade (Figure 5). On the other hand, the basic residues at the C-termini are contributed by the isoforms of the blue clade. Proline residues are conserved in both clades.

A hydrophobic motif at the N-terminus, soon after the signal peptide, is also present in the red clade (Figure 5), but with a slightly different sequence [FxxLFL(I/R)(V/D/E)Fxx(D/E)]. Moreover, we can appreciate that this first linear motif is followed by four other conserved motifs: CGGL*pp*G; (D/E)F(D/I/E)KCϕϕ(R/K); CX_5/7/9_HX_3/5/7_C; and CLY*pp*DX_3_L(Y/F/D)V. In total, five short linear motifs characterize the red clade from the N- to the C-terminus, the first one being in common with the blue clade. It would be interesting to experimentally check whether these peptides are conserved because they are antigenic or because they confer some structural features to the IDPs.

### 3.3. Nomenclature

Based on this classification and filiation, we would like to propose a more rational annotation of the gene products, trying to eliminate the gap between MEG-16 and MEG-26, and also possibly to rationalize the nomenclature of the large MEG-2 family, whose gene products have been numbered somewhat arbitrarily. It is worth mentioning that this class possesses at least nine more members deposited on UniProt, which we have excluded because they have been found only as mRNA, not yet as protein, and there is apparently no gene associated with them in WBPS.

To start a reclassification, we have taken the sequences of 13 MEG-2 proteins of the blue clade and aligned them (Figure 6), starting from the PRANK results. These proteins are coded by eight genes consecutively clustered together in the second half of chromosome 3 on the leading strand (Table 1). Only two genes not coding for MEG (*Smp_326510* and *Smp_309120*), one after the second and one before the last *meg-2*, interrupt the chain.

Interestingly, the gene coding for the red clade MEG-2/ESP15 isoform C4QPS0, *Smp_183040.1*, is located in the same part of chromosome 3, between *Smp_183010.1* and *Smp_183030.1*.

Based on ontology and genome positioning, we propose to keep the name MEG-2 to C4QPS0 of the red clade and to rename the products of the blue clade consecutively, according to their position on the genome, from 5′ to 3′, as indicated in Table 2 below.

Moreover, to disambiguate the MEG-4/antigen 10.3 proteins encoded by the genes *Smp_085840*, *Smp_307220,* and *Smp_307240*, all on chromosome 1, we propose to keep the name MEG-4.1 for the protein products of the originally deposited gene (*Smp_307220*), to call MEG-4.2 the product of the closer relative *Smp_307240* and to call *Smp_085840′*s product MEG-17 because it is more similar to MEG-12 than to MEG-4.

Analogously, to disambiguate the two proteins called MEG-8, we propose to keep the name 8 for the gene *Smp_172180.1* present on chromosome 6, given the filiation described above, and to rename the other as MEG-18.

We also propose a slight modification of MEG-10, where two sequences hold the same protein name (Table 2), as well as for the MEG-3/Grail family, which is composed of three genes coding for a total of 12 isoforms (Table 3).

An interesting case is the one of MEG-15, which in the latest annotation loses its name and becomes “uncharacterized”, therefore we propose to go back to its name, given its similarity with the MEG-2 family and with MEG-6 (see Figure 2). Indeed, the gene *Smp_010550* has four splice variants, each one coding for one isoform, unequivocally identified on UniProt. A summary is given in Table 4.

Finally, we think that some order in the MEG-1 family would also improve the readability (Table 5), although this is, together with MEG-3, one of the best-annotated families. It is better to underline that there are more than these isoforms on UniProt that have only been inferred by transcriptomics and do not (yet) belong to any deposited coding gene; therefore, they are not included in Appendix A, nor have they been used for the alignment.

### 3.4. Towards a Function?

The exact role of the MEG proteins is still unknown; their high copy number and their high variability make inferring tough. In recent years, many “omics” studies have boosted the research on schistosomes, with the aim to find new drug targets, develop more early and precise diagnostics, and implement a vaccine. A handful of studies on individual MEG proteins have been carried out, revealing their nature as intrinsically disordered proteins (IDP) without or with morphing/chameleon behavior. One morphing IDP is MEG-14, which is able to fold upon binding to negatively charged membranes or to calgranulin, a human S100 family member involved in inflammation [34,35]. MEG-24 (whose sequence was not deposited in any public repository) and MEG-27 were shown to bind to liposomes and to agglutinate red blood cells, possibly through the formation of amphipathic helices [37].

We have recently characterized by NMR three splice variants of *Smp_336990* of the MEG-2 family and confirmed their IDP nature, together with some interesting hairpin loops, which might undergo some morphing and act as a platform for interactions with host partners [38].

Indeed MEG-2 family members possess the highest content of Cys residues (Figure 5) and a conserved N-terminal motif Cys_2_X_(6/8/10)_Cys_2_, which is reminiscent of either a Zn-finger or a [2Fe2S] cluster. It remains to be proven that MEG-2 proteins might be metal sensors or chelators. A clue might be the fact that they are present in the esophagus’ secretions, where hemoglobin digestion occurs; this digestion releases high quantities of iron, which is potentially toxic. It is known that hemozoin is regurgitated [41], but maybe MEG-2 proteins could act as an iron cleaner, limiting high oxidative stress.

MEG proteins’ high variability and their high expression in the mammalian host have made the researchers think about a role in host immune evasion or modulation. This was the basis for using short synthetic peptides issued from more or less each MEG family as baits to fish IgG from infected mouse models. The most antigenic ones were then used as protective vaccines, although with low efficacy [40]. The question arises whether the full-length isoforms would be more protective or whether the conserved motifs that we have highlighted could be a better strategy.

## 4. Conclusions

The 87 verified protein products of *Schistosoma mansoni*’s 35 *micro-exon genes* (MEG) are elusive and interesting macromolecules. They are Pandora’s box of tools to understand the complex behavior of these fascinating and dangerous parasitic worms. We have presented a rationalization of the gene families based on the sequence similarities and proposed a renaming in order to avoid confusion and to help trimming what is known (the tip of the iceberg) from what still remains to be studied.

Although there are very few consensus motifs conserved in the alignment, the N-terminal signal peptide, required for secretion, remains constant among the MEG proteins. MEGs also share a prevalence of apolar residues like Pro, Phe, and Leu, and charged residues like Lys and Glu in the C-terminal region. The presence of sticky sequences, confirmed by the aliphatic index and GRAVY index calculations, suggests their potential importance in interactions with host factors. The stickiness of MEGs was also evident in studies using protein engineering for their isolation and purification. Phylogenetic analysis split the sequences into two clades and revealed distinct contributions from conserved Cys, Phe and basic residues in each clade. Furthermore, specific short linear motifs were identified within the red clade, one shared with the blue clade. These findings warrant further experimental investigation to determine whether these conserved motifs play a role in antigenicity or contribute to structural features in the IDP regions of MEGs. Overall, these insights into the characteristics and sequences of MEG proteins pave the way for future research on their functional roles and potential applications in immunization studies and interactions with host partners.

## Figures and Tables

**Figure 1 biomolecules-13-01275-f001:**
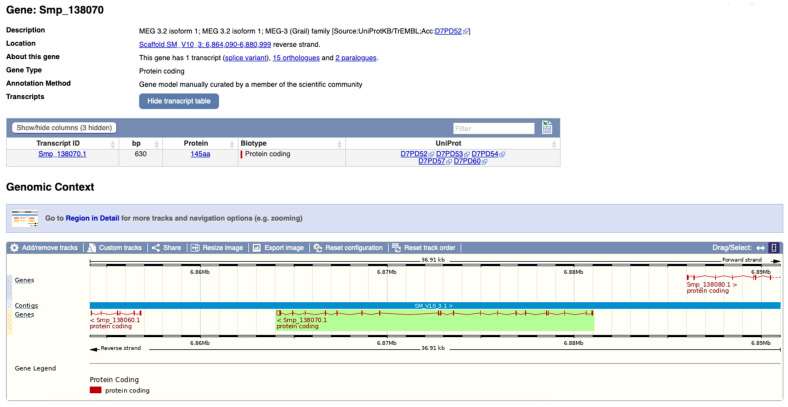
Screenshot of the entry for MEG-3.2 isoform 1 on WormBase ParaSite [12]. In the green highlighted box there is a zoom of the gene structure: vertical signs represent the exons and horizontal lines the introns. The length of each sign is directly proportional to the actual number of base pairs of exons and introns, respectively. This gene codes for 5 proteins deposited on UniProt [17].

**Figure 2 biomolecules-13-01275-f002:**
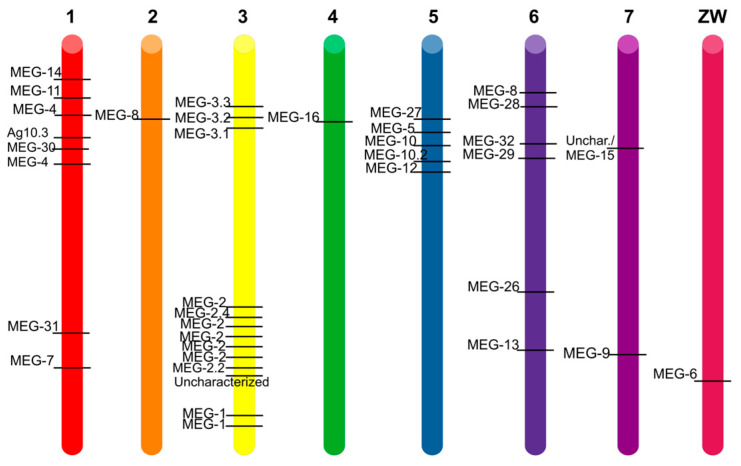
Schematic representation of *Schistosoma mansoni* haplotype. The approximate position of each MEG-coding gene on each chromosome (colored cylinder) is indicated by a black bar and its name on WormBaseParaSite is noted on the left. The chromosome number is on top of each cylinder.

**Figure 3 biomolecules-13-01275-f003:**
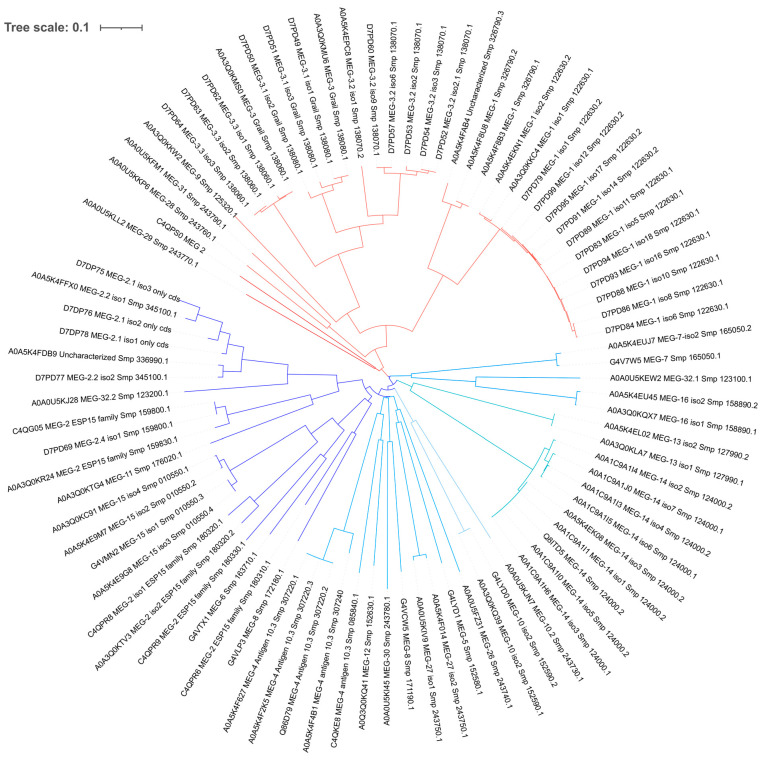
Phylogenetic tree colored in red and blue after clustering the clades by amino acid sequence similarity. Each protein is identified with its unique UniProt code, followed by its common name and WBPS gene number. In the red clade, an early event has separated MEG-29 and MEG-2 (ESP15, coded by *Smp_183040.1*) from the rest, so we have colored this branch in dark red. Similarly, on the blue clade, MEG-7, MEG-32 and MEG-16 departed early from the clade and are highlighted in light blue.

**Figure 4 biomolecules-13-01275-f004:**
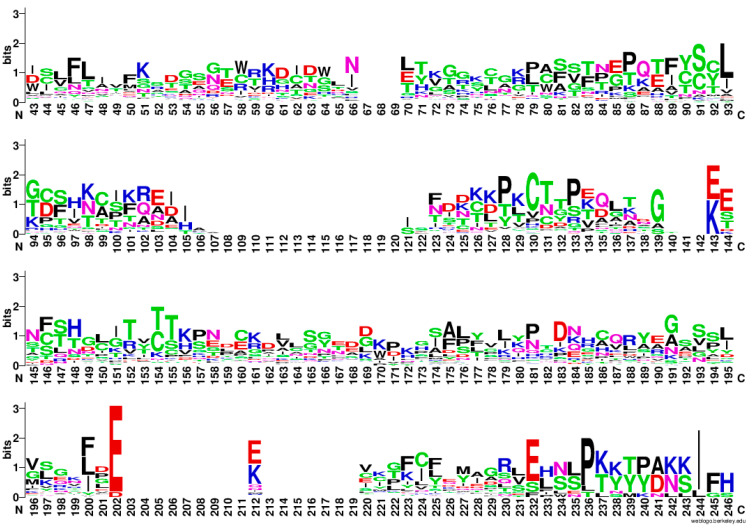
WebLogo representation of the alignment of all the 87 MEG protein sequences. The sequences of the signal peptide have been omitted for clarity. Even if the longest protein is 189 residues long, the number of gaps lengthens the aligned sequences to 246 residues.

**Figure 5 biomolecules-13-01275-f005:**
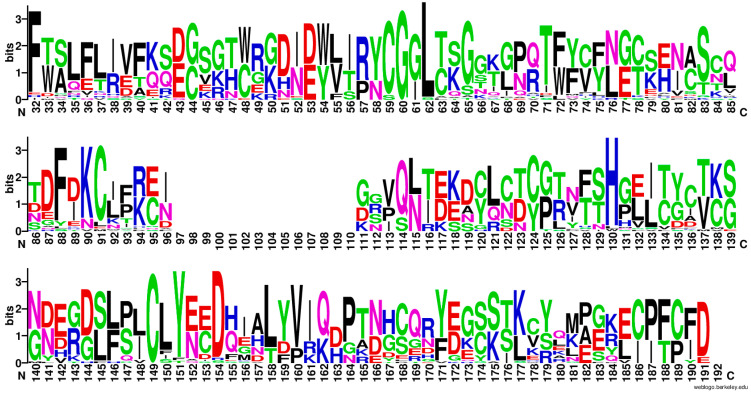
WebLogo representation of the alignment of the red clade composed of 35 MEG proteins, i.e., isoforms of MEG-1, MEG-3, MEG-9, MEG-28, MEG-29, MEG-31, and C4QPS0 of the MEG-2 family. The sequence of the signal peptide has been omitted for clarity.

**Figure 6 biomolecules-13-01275-f006:**
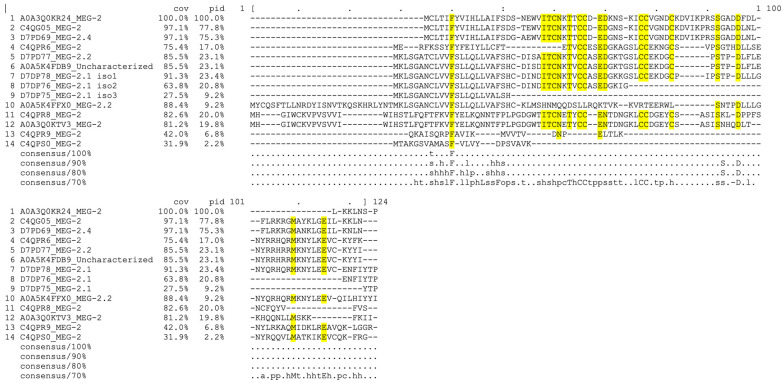
Sequence alignment of proteins coded by the *meg-2* family. Sequences from 1 to 13 belong to the blue clade and the latter sequence (#14) belongs to the red clade. Conserved residues are highlighted in yellow.

**Table 1 biomolecules-13-01275-t001:** Summary of the cluster of the **blue** components of *meg-2* on chromosome 3.

	MEG-2 Coding Genes from 5′ to 3′ in the Second Half of Chromosome 3						
**WBPS gene identifier**	*Smp_1598030.1*	*Smp_159800.1*	*Smp_183010.1*	*Smp_183030.1*	*Smp_183020.1* and *Smp_183020.2*	*Smp_345100.1*	*Smp_336990*
**UniProt ID**	A0A3Q0KR24	C4QG05 and D7PD69	C4QPR6	C4QPR9	C4QPR8 and A0A3Q0KTV3	A0A5K4FFX0 and D7PD77	A0A5K4FDB9 D7DP78 D7DP76 D7DP75

**Table 2 biomolecules-13-01275-t002:** Proposed changes in the nomenclature of MEG protein products -2, -4, -8, -10 based on the filiation presented in Figure 3.

WBPS Gene ID	UniProt ID	Old Name	New Name
*Smp_183040.1*	C4QPS0	MEG-2/ESP15	MEG-2
*Smp_1598030.1*	A0A3Q0KR24	MEG-2/ESP15	MEG-2.1
*Smp_159800.1*	C4QG05	MEG-2/ESP15	MEG-2.2 isoform 1
	D7PD69	MEG-2.4 isoform 1	MEG-2.2 isoform 2
*Smp_183010.1*	C4QPR6	MEG-2/ESP15	MEG-2.3
*Smp_183030.1*	C4QPR9	MEG-2/ESP15	MEG-2.4
*Smp_183020.1*	C4QPR8	MEG-2 isoform 1	MEG-2.5 isoform 1
*Smp_183020.2*	A0A3Q0KTV3	MEG-2 isoform 2	MEG-2.5 isoform 2
*Smp_345100.1*	A0A5K4FFX0	MEG-2.2 isoform 1	MEG-2.6 isoform 1
	D7PD77	MEG-2.2 isoform 2	MEG-2.6 isoform 2
*Smp_336990*	A0A5K4FDB9	Uncharacterized	MEG-2.7 isoform 1
	D7DP78	MEG-2.1 isoform 1	MEG-2.7 isoform 2
	D7DP76	MEG-2.1 isoform 2	MEG-2.7 isoform 3
	D7DP75	MEG-2.1 isoform 3	MEG-2.7 isoform 4
*Smp_307220.1*	A0A5K4F627	Developmentally regulated antigen 10.3	MEG-4.1 isoform 1
*Smp_307220.2*	A0A5K4F2K5	Developmentally regulated antigen 10.3	MEG-4.1 isoform 2
*Smp_307220.3*	Q86D79	Developmentally regulated antigen 10.3	MEG-4.1 isoform 3
*Smp_307240.1*	A0A5K4F4B1	MEG-4/antigen 10.3	MEG-4.2
*Smp_085840*	C4KE8	MEG-4/antigen 10.3	MEG-17
*Smp_172180.1*	G4VLP3	MEG-8	MEG-8
*Smp_171190.1*	G4VCW5	MEG-8	MEG-18
*Smp_152590.1*	A0A3Q0KQ39	MEG-10 isoform 2	MEG-10.1 isoform 1
*Smp_152590.2*	G4LYD0	MEG-10 isoform 2	MEG-10.1 isoform 2
*Smp_243730.1*	A0A0U5KJN7	MEG-10.2	MEG-10.2

**Table 3 biomolecules-13-01275-t003:** Proposed changes in the nomenclature of MEG-3 protein products, based on the filiation presented in Figure 3.

WBPS Gene ID	UniProt ID	Old Name	New Name
*Smp_138060.1*	D7PD62	MEG-3.3 isoform 1	
	D7PD63	MEG-3.3 isoform 2	
	D7PD64	MEG-3.3 isoform 3	
	A0A3Q0KMS0	MEG-3 Grail family	MEG-3.3 isoform 4
*Smp_138070.1*	D7PD52	MEG-3.2 isoform 2.1	MEG-3.2 isoform 1
	D7PD53	MEG-3.2 isoform 2	
	D7PD54	MEG-3.2 isoform 3	
	D7PD57	MEG-3.2 isoform 6	MEG-3.2 isoform 4
	D7PD60	MEG-3.2 isoform 9	MEG-3.2 isoform 5
*Smp_138080.1*	D7PD49	MEG-3.1 isoform 1	
	D7PD51	MEG-3.1 isoform 3	MEG-3.1 isoform 2
	A0A3Q0KMU6	MEG-3 Grail family	MEG-3.1 isoform 3

**Table 4 biomolecules-13-01275-t004:** Proposed changes in the nomenclature of MEG-15 protein products.

WBPS Gene ID	UniProt ID	Old Name	New Name
*Smp_010550.1*	A0A3Q0KC91	Uncharacterized protein	MEG-15 isoform 1
*Smp_010550.2*	A0A5K4E9M7	Uncharacterized protein	MEG-15 isoform 2
*Smp_010550.3*	G4VMN2	Uncharacterized protein	MEG-15 isoform 3
*Smp_010550.4*	A0A5K4E9G8	Uncharacterized protein	MEG-15 isoform 4

**Table 5 biomolecules-13-01275-t005:** Proposed changes in the nomenclature of MEG-1 protein products.

WBPS Gene ID	UniProt ID	Old Name	New Name
*Smp_326790.1*	A0A5K4F8B3	MEG-1	MEG-1.1 isoform 1
	A0A5K4F8U8	MEG-1	MEG-1.1 isoform 2
*Smp_122630.1*	A0A3Q0KKC4	MEG-1 isoform 1	MEG-1.2 isoform 1
	D7PD83	MEG-1 isoform 5	MEG-1.2 isoform 2
	D7PD84	MEG-1 isoform 6	MEG-1.2 isoform 3
	D7PD86	MEG-1 isoform 8	MEG-1.2 isoform 4
	D7PD88	MEG-1 isoform 10	MEG-1.2 isoform 5
	D7PD89	MEG-1 isoform 11	MEG-1.2 isoform 6
	D7PD93	MEG-1 isoform 16	MEG-1.2 isoform 7
*Smp_122630.2*	A0A5K4EKN1	MEG-1 isoform 2	MEG-1.2 isoform 8
	D7PD79	MEG-1 isoform 1	MEG-1.2 isoform 9
	D7PD91	MEG-1 isoform 14	MEG-1.2 isoform 10
	D7PD94	MEG-1 isoform 18	MEG-1.2 isoform 11
	D7PD95	MEG-1 isoform 17	MEG-1.2 isoform 12
	D7PD99	MEG-1 isoform 12	MEG-1.2 isoform 13

## Data Availability

No new data were created. All the data used in the manuscript are freely available on the WBPS and UniProt databases.

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
