# Peer review of "Revisiting Schistosoma mansoni Micro-Exon Gene (MEG) Protein Family: A Tour into Conserved Motifs and Annotation"

_biomolecules, 2023, doi:10.3390/biom13091275_

Round 1

Reviewer 1 Report

The study provides deeper insights in a gene family of unknown function in a human parasite. No new data is produced and therefore I am not sure if the study can be characterised as an article or meta analysis or another category. Nevertheless, the study is well structured presenting an in depth analysis.

I have a major concern: why did the authors only use WormBase and not GenBank? Please explain. And if possible please add in the phylogenetic tree the GenBank accession number of each sequence.

Also, although figures are indicative, a schematic depiction of the general annotation of the genes MEG genes, with different lengths for introns and axons accordingly, should be prepared and added to be easier for the reader to understand the annotation.

Author Response

We would like to express our gratitude to both reviewers who made constructive comments to improve our manuscript.

Reviewer 1

"The study provides deeper insights in a gene family of unknown function in a human parasite. No new data is produced and therefore I am not sure if the study can be characterised as an article or meta analysis or another category. Nevertheless, the study is well structured presenting an in depth analysis."

Indeed this reviewer is correct in the sense that we did not deposit new sequences of the MEG superfamily; however, we have performed a thorough recollection and meta-analysis of those curated sequences, which have been deposited in WormBaseParaSite (WBPS), a comprehensive database dedicated to helminth genomics.

"I have a major concern: why did the authors only use WormBase and not GenBank? Please explain. And if possible please add in the phylogenetic tree the GenBank accession number of each sequence."

We have intentionally used only the identifiers from WBPS because this is a reference for helminth genomics. Moreover, this DB belongs to the ENSEMBL Genomes programme, an integrative resource for genome-scale data from non-vertebrate species. Most of the first identified MEG sequences had been directly deposited in Genbank from the NCBI, before ENSEMBL started, but lately the “schisto” community has agreed in using and updating only the WBPS entries. We thank the reviewer for their suggestion, but we think that adding this other identifier to an already packed phylogenetic tree will reduce the readability of the figure. We have therefore decided to add the GenBank ID to the comprehensive table in the supplementary material.

"Also, although figures are indicative, a schematic depiction of the general annotation of the genes MEG genes, with different lengths for introns and axons accordingly, should be prepared and added to be easier for the reader to understand the annotation."

We are quite puzzled by this comment, which maybe we have not entirely understood. This is the scope of the full annotation of WBPS and, in fact, figure 1 shows exactly one of these entries with the length of the introns and exons. We have therefore modified the legend of this figure, to better specify and possibly make the point clearer. Regarding the proposed annotation, we decided to be the most conservative as possible, hence there is no relation between the isoform number and the length of the protein. The original consecutive numbers reflected the timeline of their discovery and the relative quantity of proteins found in the proteomics studies. We have added on those and proposed a disentangling based on the vicinity of the phylogenetic tree. This is based on protein sequence similarity and not on the length of the exons.

Reviewer 2 Report

The authors aim at deciphering the phylogenetic relationships og meg genes in Schistosoma mansoni, the causative agent of an important human parasitoses. By multiple sequence alignment they highlight the filiation of meg gene from a putative common ancestor, likely resideing on extant ch. 6 , which later spreaded into at least two main family. The paper sound interesting in view of the possibility to study each single MEG protein as antigenic target to develop vaccines against schistosomiasis. Therefore, the paper should be considered of leading interest for a specialized audience. Nevertheless, the proposed new onthology og meg genes is important for sake of clairty, sinca conflicting naming is present in literature. 

My complaints regard only minor typos in english editing (please, see attahced file with yellow highlighting). In addition, I suggest here two improvements for sake of exhaustiveness and clarity:

1. since authors report literature about studies on some MEG proteins (like MEG3.2, MEG3.4, MEG-4 and some others), does it possible to predict the structure of these ones, for example by using tools like Modeller, iTASSER or alphaFold? This could be valuable in view of the antigenic potential of MEGs (see line 234-235). Results about these predictions might add values to the already present primary sequences' studies and foster or add clues on what already reported in the section 3.4.

2. in fig. 3, it is hard to distinguish the different shades of colour, thus my suggestion is to colour the genes' names, otherwise to add a coloured box around the genes' names.

English are fine, few minor issues should be checked (highlighted in the attached file)

Author Response

We would like to express our gratitude to both reviewers who made constructive comments to improve our manuscript. Here are our point-by-point answers.

“The authors aim at deciphering the phylogenetic relationships og meg genes in Schistosoma mansoni, the causative agent of an important human parasitoses. By multiple sequence alignment they highlight the filiation of meg gene from a putative common ancestor, likely resideing on extant ch. 6 , which later spreaded into at least two main family. The paper sound interesting in view of the possibility to study each single MEG protein as antigenic target to develop vaccines against schistosomiasis. Therefore, the paper should be considered of leading interest for a specialized audience. Nevertheless, the proposed new onthology og meg genes is important for sake of clairty, sinca conflicting naming is present in literature.”

We appreciate a lot the encouraging comments of the reviewer, who has understood the importance of disentangling the nomenclature, whenever an in-depth study needs to be carried out on MEGs role on the host-parasite interactions.

“My complaints regard only minor typos in english editing (please, see attahced file with yellow highlighting). In addition, I suggest here two improvements for sake of exhaustiveness and clarity:

1. since authors report literature about studies on some MEG proteins (like MEG3.2, MEG3.4, MEG-4 and some others), does it possible to predict the structure of these ones, for example by using tools like Modeller, iTASSER or alphaFold? This could be valuable in view of the antigenic potential of MEGs (see line 234-235). Results about these predictions might add values to the already present primary sequences' studies and foster or add clues on what already reported in the section 3.4.”

Indeed, we have modified the typos accordingly, thank you.

As for the models, at present each one of the entries on UniProt DB contains a model from AlphaFold of ditto MEG. The probability score is, however, quite low, except for the signal peptide, since there are no homologous proteins of known structure. Modeller and iTASSER, as well, give very low confidence models for the reasons above and also because MEGs are intrinsically disordered proteins (IDPs). Therefore, we believe that such figures will not add useful data to the manuscript; on the other hand, a thorough analysis of the conserved motifs, which we have put forward for the first time, is indeed more informative for rationalizing the role of MEGs in host-parasite interactions.

“2. in fig. 3, it is hard to distinguish the different shades of colour, thus my suggestion is to colour the genes' names, otherwise to add a coloured box around the genes' names.”

We thank the reviewer and agree about the readability of Figure 3, thus we have added the boxes on the names, as suggested.